# CO$_2$ Hydrogenation over Fe-Co Bimetallic Catalyst Derived from the Thermolysis of [Co(NH$_3$)$_6$][Fe(CN)$_6$]

**Alevtina N. Gosteva** [1,*], **Mayya V. Kulikova** [2], **Mikhail I. Ivantsov** [2], **Alena A. Grabchak** [2], **Yulya P. Semushina** [1], **Semen E. Lapuk** [3], **Alexander V. Gerasimov** [3] **and Nikita S. Tsvetov** [1]

1   I.V. Tananaev Institute of Chemistry—Subdivision of the Federal Research Centre «Kola Science Centre of the Russian Academy of Sciences» Akademgorodok, 26a, 184209 Apatity, Russia; semushinajp@mail.ru (Y.P.S.); tsvet.nik@mail.ru (N.S.T.)
2   A.V. Topchiev Institute of Petrochemical Synthesis, RAS, Leninsky Prospect, 29, 119991 Moscow, Russia; m_kulikova@ips.ac.ru (M.V.K.); ivantsov@ips.ac.ru (M.I.I.); ale.grabchak@ips.ac.ru (A.A.G.)
3   Department of Physical Chemistry, A.M. Butlerov Institute of Chemistry, Kazan Federal University, Kremlevskaya, 18, 420008 Kazan, Russia; lapuksemen@gmail.com (S.E.L.); alexander.gerasimov@kpfu.ru (A.V.G.)
*   Correspondence: angosteva@list.ru

**Abstract:** Reducing the amount of CO$_2$ in the atmosphere is a very important task. Therefore, the development and search for new approaches to the synthesis of catalytic systems, allowing for the catalytic conversion of CO$_2$ into valuable products, is an urgent task. In this work, the catalyst was obtained by the thermolysis of a double complex compound. In this regard, kinetic studies of the parameters of the thermolysis process of double complex salts-[Co(NH$_3$)$_6$][Fe(CN)$_6$] were additionally determined using isoconversion and model approaches of non-isothermal kinetics. The catalyst was studied using various physicochemical methods—X-ray diffraction (XRD), infrared (IR)-spectroscopy, Raman spectroscopy, and X-ray photoelectron spectroscopy (XPS). It was shown that, at the stage of catalyst preparation, the formation of a CoFe alloy occurred, while the surface mainly consisted of carbon in sp$_2$-hybridization, and the metals existed in the form of spinel CoFe$_2$O$_4$. It was shown that catalysts based on bimetallic salts were active in the process of hydrogenation of carbon dioxide without a pre-activation stage (CO$_2$ conversion reached 28%, with a specific activity of 4.0 μmolCO$_2$/gMe·s). It was established that it was possible to change the selectivity of the carbon dioxide hydrogenation process by pre-treating the catalyst with hydrogen (selectivity for methane formation in the presence of an unreduced catalyst is 46.4–68.0%, whereas in the presence of a reduced catalyst it is 5.1–16.5%).

**Keywords:** CO$_2$ hydrogenation; catalysis; double complex compound; thermokinetics; thermal decomposition

## 1. Introduction

Currently, processes aimed at limiting the impact of greenhouse gas emissions, including carbon dioxide, on the climate are relevant. One of the options for involving CO$_2$ in chemical transformations is its hydrogenation to produce synthetic hydrocarbons and oxygenates. In the mid-2000s, one of the common areas of research was the development of technologies for the chemical utilization of carbon dioxide aimed at converting it into valuable products [1–5]. In particular, researchers paid considerable attention to the conversion of CO$_2$ into synthetic liquid hydrocarbons.

There are various ways of converting CO$_2$ into valuable chemical products, which include electrochemical and photoelectrochemical processes, heterogeneous catalysis, etc. [6,7].

The process of the chemical transformation of $CO_2$ into various chemical products can be divided into two chemical reactions:

$$CO_2 + H_2 = CO + H_2O$$

$$CO + H_2 = -CH_2- + H_2O$$

The implementation of these processes occurs at various active centers, and for effective implementation it is necessary to provide access to two centers.

Most of the research that has been carried out in this area is devoted to the one-stage synthesis of hydrocarbons using bifunctional catalysts.

The use of iron and cobalt catalysts are two main research directions in the development of processes for the catalytic hydrogenation of carbon oxides. Various types of active sites are formed on the surface of catalysts. The idea of combining active centers formed by iron and cobalt through the composition of an iron–cobalt catalyst was implemented in [8].

Many studies [9–18] have examined the activity of iron, cobalt, and iron–cobalt catalysts. These studies have shown that, although the individual metals Co and Fe are also active, the best activity is achieved with Co-Fe alloys.

For most catalytic systems, the active phase is formed at the stage of preliminary activation, which involves treatment with hydrogen or special reagents at a fairly high temperature (400–450 °C).

There are a large number of methods for synthesizing functional materials. One such method is the thermal decomposition of complex compounds. This method has the following advantages: (1) a relatively short production time, (2) a relatively low environmental impact, and (3) the use of simple equipment. Double complexes (DCS) can be used to obtain the above synergistic effect from the simultaneous use of cobalt and iron. Stoichiometry is the main advantage of using DCS. It determines the stoichiometry of the final product and makes it possible to determine its composition at the stage of precursor synthesis.

In the course of this work, the authors propose to use $[Co(NH_3)_6][Fe(CN)_6]$ as a precursor for $CO_2$ hydrogenation. The thermolytic product of this compound has previously shown good catalytic properties [19]. CO conversion reaches 90%. The maximum distribution of hydrocarbons falls on hydrocarbons with a chain length of 6–8 carbon atoms.

The main method of catalyst preparation is thermal decomposition in this study. Therefore, it is important to consider the process of thermolysis of $[Co(NH_3)_6][Fe(CN)_6]$ in more detail. Only a description of the process of dynamic (synchronous thermal analysis) and static thermal decomposition of this binary complex can be found in the literature. The thermogravimetric analysis with infrared spectroscopy (TG–IR) study of the DCS in a nitrogen flow was carried out by [20]. Two stages of weight loss are observed on the differential thermogravimetric (DTG)curve: the first from 190 to 250 °C and the second from 600 to 620 °C. The minimum on the DTG curve at 200 °C coincides with the maximum release of ammonia (it is released in the region of 190–420 °C). The second fixed gaseous product is HCN. Its release is observed in the region of 190–500 °C, with a maximum of ~320 °C. The second mass loss peak at ~600 °C is not accompanied by the release of any gaseous product. According to the authors of the article [20], the release of $N_2$ occurs, which cannot be fixed in a stream of nitrogen.

The structure of catalytic systems—their composition and the symmetry of the components—strongly influences the activity and selectivity of the catalyzed processes; these influences are especially observed in the processes of the photocatalytic and electrocatalytic conversion of carbon dioxide [21–24]. However, the influence of the structure of heterogeneous catalysts has also been widely studied [1]. The use of bimetallic complex compounds as catalyst precursors will make it possible to synthesize a catalyst, in the structure of which there will be an effective uniform mixing of Co and Fe, since they are present in one complex molecule.

The goal of this work was to determine the nature of the kinetic processes occurring during the thermolysis of $[Co(NH_3)_6][Fe(CN)_6]$. Studying the thermal decomposition kinetics of hydrogenation catalysts is necessary to understand catalyst characteristics, elucidate

reaction mechanisms, and evaluate catalyst stability. By studying the decomposition rates of hydrogenation catalysts under different conditions, their catalytic activity and selectivity can be better understood. The study of thermal decomposition kinetics contributes to the improvement in hydrogenation processes, allowing the development of efficient and stable catalysts for various industrial applications. The thermolytic product of this DCS was tested as a catalyst for the $CO_2$ hydrogenation reaction. The influence of the presence of activation on the catalytic process was studied.

## 2. Results and Discussion

### 2.1. Thermokinetic Study of [Co(NH₃)₆][Fe(CN)₆]

To determine the kinetic parameters of the $[Co(NH_3)_6][Fe(CN)_6]$ thermolysis process, according to the International Confederation for Thermal Analysis and Calorimetry(ICTAC)recommendations [7–9], three non-isothermal scans were performed. The $[Co(NH_3)_6][Fe(CN)_6]$ sample was heated at rates of 1, 10, and 20 °C/min. The temperature range was from 40 to 1000 °C. TG heating curves with different rates are shown in Figure 1.

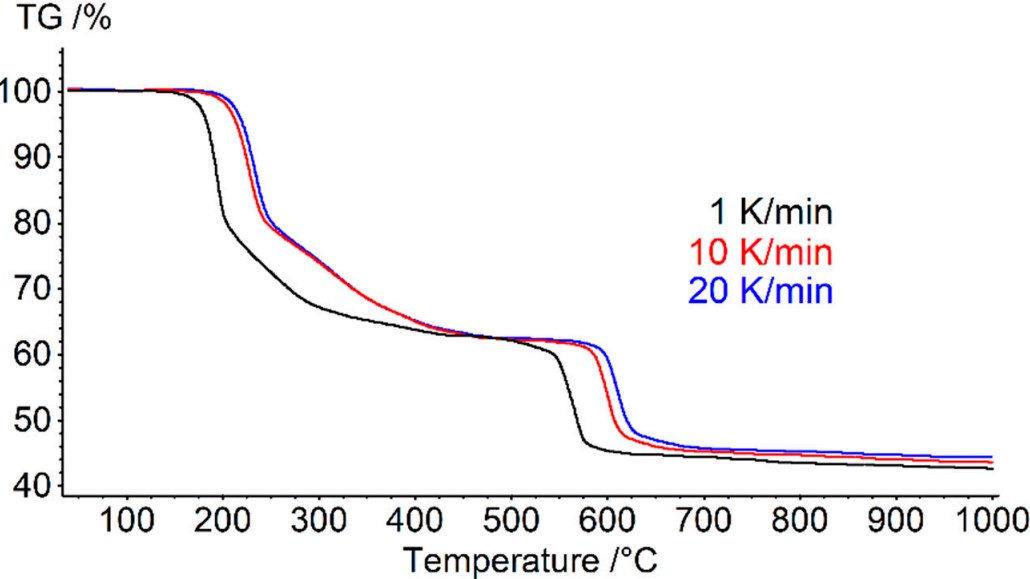

**Figure 1.** TG heating curves of $[Co(NH_3)_6][Fe(CN)_6]$ at rates of 1, 10, and 20 °C/min. All studies were carried out in an argon atmosphere with a flow rate of 75 mL/min.

In the temperature range of 40–500 °C, a mass loss of ~38% was observed, corresponding to the release of $NH_3$ and HCN from the complex [20]. At temperatures > 500 °C, a mass loss of ~17% was observed, which may be associated with the release of molecular nitrogen [20]. The total weight loss was 55%.

As can be seen from Figure 1, the thermolysis of $[Co(NH_3)_6][Fe(CN)_6]$ consisted of two main steps. The first stage of thermolysis occurred in the temperature range from 100 to 500 °C. The second stage occurred above 500 °C.

Using isoconversion approaches (Friedman analysis and Kissinger–Akahira–Sunose (KAS) method), the activation energies and pre-exponential factors of the first and second stages of thermolysis of $[Co(NH_3)_6][Fe(CN)_6]$ were determined, depending on the degree of conversion ($\alpha$). Figure 2 shows the dependences of the activation energy and the pre-exponential factor of the Arrhenius equation $\alpha$ obtained by using the Friedman analysis and KAS methods for the first stage of $[Co(NH_3)_6][Fe(CN)_6]$ thermolysis.

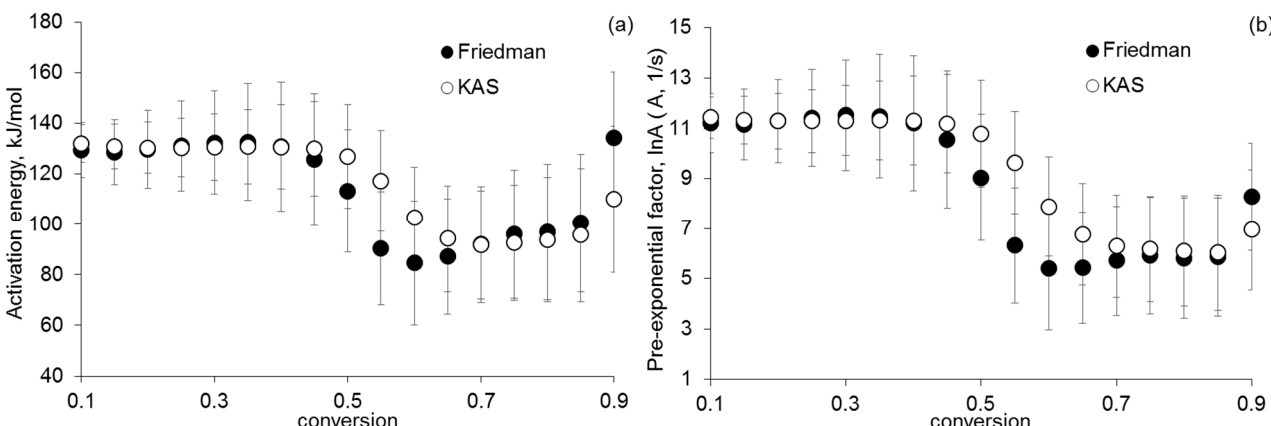

**Figure 2.** Dependence of (**a**) activation energy and (**b**) pre-exponential factor $\alpha$, found by using the Friedman analysis and the KAS method for the first stage of thermolysis.

According to the data in Figure 2, the kinetic parameters obtained by using isoconversion approaches have similar values. The dependences of the activation energy on $\alpha$ have a sharp inflection in the region where $\alpha \sim 0.5$. This may indicate two successive reactions at the first stage of thermolysis. The first reaction occurs at $\alpha < 0.5$, and the second one occurs at $\alpha > 0.5$. The average value of the activation energy and the pre-exponential factor of the first reaction is 130 kJ/mol and 11 s$^{-1}$, respectively. Figure 3 shows the kinetic parameters determined by using isoconversion approaches of non-isothermal kinetics for the second stage of $[Co(NH_3)_6][Fe(CN)_6]$ thermolysis.

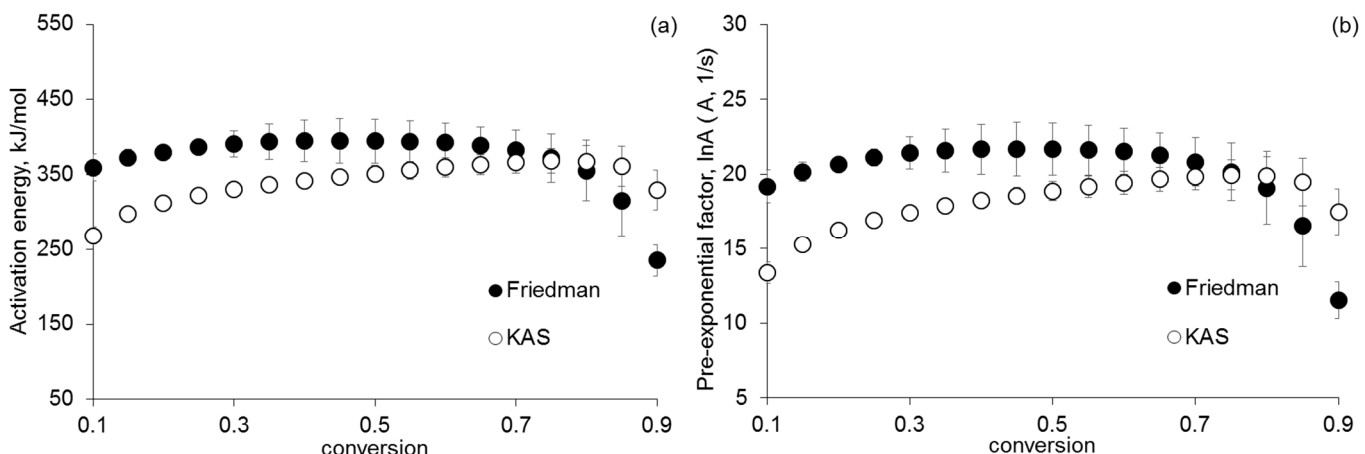

**Figure 3.** Dependence of (**a**) activation energy and (**b**) pre-exponential factor on $\alpha$, found by using the Friedman analysis and the KAS method for the second stage of thermolysis.

Figure 3 shows that the values of the kinetic parameters of the second stage of thermolysis increase monotonically. The average values of the activation energy and the pre-exponential factor determined by using the Friedman analysis are 370 kJ/mol and 20 s$^{-1}$, respectively. The corresponding values determined using the KAS method are 320 kJ/mol and 17 s$^{-1}$.

Model methods of non-isothermal kinetics were used to determine the kinetic parameters of each stage of the $[Co(NH_3)_6][Fe(CN)_6]$ thermolysis. According to the kinetic data obtained by using isoconversion approaches, the first thermolysis step was divided into two consecutive reactions. Kinetic parameters of the thermolysis (coefficient of determination and F-test values) are presented in Tables S1 and S2. Models with the highest coefficient of determination are presented in Table 1.

**Table 1.** Kinetic parameters of the $[Co(NH_3)_6][Fe(CN)_6]$ thermolysis determined by using model methods.

| The First Stage | | The Second Stage |
| --- | --- | --- |
| The First Reaction | The Second Reaction | |
| Cnm | Sb | Cnm |
| E = 127.3 kJ/mol | E = 77.6 kJ/mol | E = 345.7 kJ/mol |
| logA = 11.1 (A, s$^{-1}$) | logA = 4.8 (A, s$^{-1}$) | logA = 18.2 (A, s$^{-1}$) |
| Reactorder n = 1.2 | Reactorder n = 3.7 | Reactorder n = 2.8 |
| Log(AutocatPreExp) 0.4 | Autocat Order 0.4 | Log(AutocatPreExp) 1.6 |
| Autocat Power m = 0.8 | LogOrder q = 0.8 | Autocat Power m = 1.7 |
| R$^2$ | | |
| 0.99957 | | 0.99928 |

The kinetic parameters obtained by using model methods are in agreement with the parameters determined by using isoconversion approaches of non-isothermal kinetics. This indicates the reliability of the obtained results. It should be noted that all phases of the first and second stages of thermolysis are well described by autocatalytic models. This may indirectly indicate the catalytic mechanism of the first and second stages of $[Co(NH_3)_6][Fe(CN)_6]$ thermolysis.

*2.2. Catalytic Tests*

Based on the literature data [20] and our above study of the thermokinetic behavior of the DCS, it follows that the thermal decomposition of the DCS $[Co(NH_3)_6][Fe(CN)_6]$ and the removal of the gas phase completely ends by 650 °C. Therefore, the study of catalytic activity was carried out for the catalyst obtained by thermolysis at this temperature. Using physicochemical methods of analysis, it was determined that the catalyst is a CoFe alloy with a carbon content of 25.4 wt.%, a Co content of 35.8 wt.%, and Fe content of 32.1 wt.%. The sample has a specific surface area of 36.4 m$^2$/g. The pores are slit-like, and the average pore size is ~40 nm.

The IR spectra of the catalytic composition before activation are shown in Figure 4. The spectrum of potassium bromide is given for comparison. It was this KBr that was used for the analysis. It can be seen that the spectra are almost identical. The only peak at 2143 cm$^{-1}$ is directly related to the catalyst and not to KBr. The peak can be attributed to the vibration of the bond of Fe bridge–CN [25]. But this fluctuation has insignificant intensity. Therefore, it can be assumed that undecomposed cyano groups are found in trace amounts.

This is also confirmed by the elemental analysis of the sample for nitrogen, both before and after its activation with hydrogen, the nitrogen total of which is 0.94 and 0.68%, respectively.

Catalytic tests were carried out in two modes: without the activation stage and with the activation stage. In the process of hydrogenation of carbon dioxide without prior activation, increasing the temperature increases the conversion of carbon dioxide. The selectivity for the formation of $CH_4$ decreases with an increasing temperature from 68.0 to 46.4%. Dependencies for selectivity for $C_2$–$C_4$ and $C_{5+}$ hydrocarbons are inverse; selectivity indicators increase with an increasing temperature up to 16.8% ($C_2$–$C_4$) and up to 20.9% for $C_{5+}$. The maximum selectivity for CO is achieved at 270 °C—16.8% (Table 2). The specific activity–metal time yield (MTY) is also presented in Table 2.

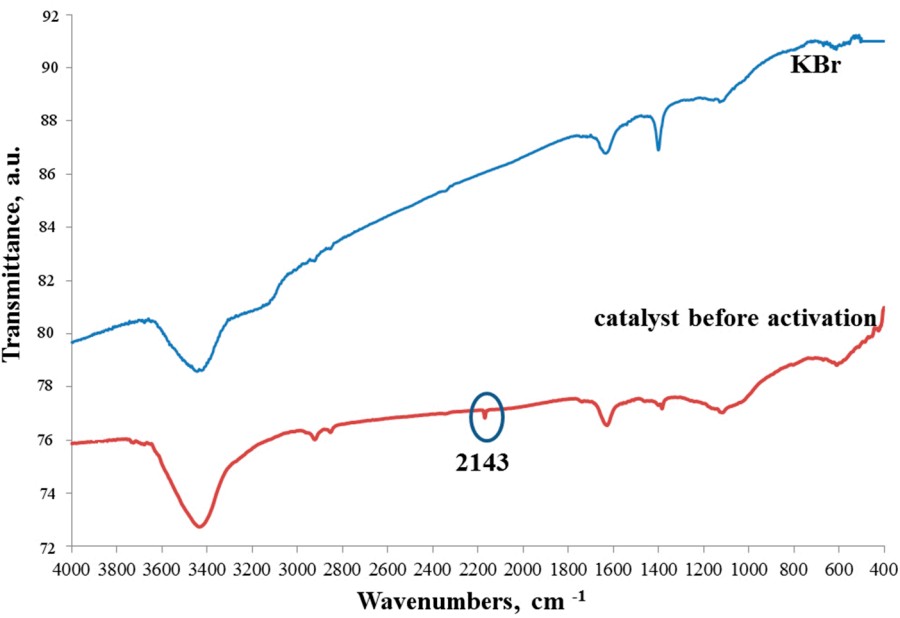

**Figure 4.** IR spectra of the catalyst before activation and KBr.

**Table 2.** The selectivity and $CO_2$ conversion of the process of hydrogenation of carbon dioxide.

| T, °C | $X_{CO_2}$, % | Selectivity, % | | | | $MTY \times 10 \times 10^6$, $mole_{CO_2}/g_{Me} \times s$ |
|---|---|---|---|---|---|---|
| | | $C_1$ | $C_2$–$C_4$ | $C_{5+}$ | CO | |
| Without activation stage (WAS) | | | | | | |
| 230 | 3.5 | 68.0 | 9.3 | 8.1 | 14.6 | 1.8 |
| 250 | 8.7 | 59.0 | 11.1 | 15.6 | 14.2 | 1.4 |
| 270 | 15.2 | 52.7 | 14.5 | 16.0 | 16.8 | 2.5 |
| 290 | 22.3 | 50.8 | 15.7 | 19.6 | 14.0 | 3.7 |
| 310 | 28.0 | 46.4 | 16.8 | 20.9 | 15.9 | 4.0 |
| Activation stage (AS) | | | | | | |
| 230 | 11.6 | 5.1 | 1.7 | 78.3 | 14.8 | 2.3 |
| 250 | 14.0 | 6.8 | 4.2 | 71.8 | 17.3 | 2.8 |
| 270 | 17.9 | 10.8 | 2.2 | 63.7 | 23.4 | 3.5 |
| 290 | 20.5 | 14.2 | 3.5 | 55.6 | 26.7 | 4.0 |
| 310 | 26.3 | 16.6 | 4.4 | 53.9 | 25.1 | 5.2 |

The second mode of catalytic tests was an experiment with the activation stage, which consisted of an hourly treatment of the catalyst with hydrogen. Due to activation, the initial conversion of carbon dioxide is much higher than that of the non-activated sample and starts from 11.6%, but the increase in conversion is not so intense: at 310 °C, the values of $CO_2$ conversion are almost same for the activated and non-activated sample—26–28%. However, the selectivity of methane and light hydrocarbons $C_2$–$C_4$ is low; during the experiment they do not exceed 16.6 and 4.4%, respectively. The selectivity for the formation of $C_{5+}$ decreases with an increasing temperature from 78.3 to 53.9%, which significantly exceeds the selectivity of the corresponding product at the same temperatures in the case of synthesis without activation. Also, a higher selectivity for CO is achieved at 290 °C—26.7%.

In the low-temperature region (up to 270 °C), the hydrogenating activity of $CO_2$ upon activation is somewhat higher, but with an increase in the temperature of catalysis with preliminary activation, it is inferior in activity to catalysis without activation. The overall selectivity of WAS for light hydrocarbons ($C_1$–$C_4$) is comparable to the selectivity of AS for

$C_{5+}$ hydrocarbons. The situation is similar with the selectivity of $C_1$–$C_4$ AS and $C_{5+}$ WAS. As can be seen from the presented data, the catalyst obtained by the thermolysis of DCS is not inferior in activity to bimetallic catalysts for $CO_2$ hydrogenation, Table 3.

**Table 3.** Comparison of the main indicators of the carbon dioxide hydrogenation process with literature data.

| Catalyst | Ratio $CO_2$:$H_2$ | Reaction Conditions | Activation Stage | Conversion $CO_2$ % | Selectivity % | | | | |
|---|---|---|---|---|---|---|---|---|---|
| | | | | | $C_1$ | $C_2$–$C_4$ | $C_{5+}$ | CO | |
| 5K-10Co/Fe | 2.65:1 | 300 °C 1.0 MPa (GHSV) 12 g·h/mole | - | 29.4 | 13.2 | 36.3 | 50.5 | 22.5 | [8] |
| Fe-Co(0.5)/Al$_2$O$_3$ | 3:1 | 300 °C 1.1 MPa | H$_2$ 400 °C 2 h. | 33.1 | 87.0 | 12.0 | | 1.0 | [9] |
| Fe–Co(0.50)/K(0.3)/Al$_2$O$_3$ | | | | 50.3 | 63.0 | 36.0 | | 1.0 | |
| CoFe-3.54Na | 3:1 | 240 °C, 3 MPa (GHSV) 5500 mL·g$^{-1}$·h$^{-1}$ | - | 7.0 | 12.7 | 14.9 | 72.3 | 8.8 | [14] |
| CoFe-0.81Na | | | | 10.2 | 17.8 | 9.4 | 73.9 | 5.2 | |
| CoFe-0.23Na | | | | 12.6 | 55.1 | 5.4 | 39.5 | 5.0 | |
| CoFe | | | | 19.6 | 70.3 | 2.3 | 27.4 | 2.9 | |
| Fe-Co-K | 3:1 | 320 °C, 20 MPa, (GHSV) 75 mL/min | H$_2$ 400 °C 5 h | 18.7 | 38.0 | 4.8 | 15.5 | 69.0 | [16] |
| 20Fe-Co-K/80MMC | | | | 21.5 | 36.0 | 3.2 | 11.4 | 65.0 | |
| 40Fe-Co-K/60MMC | | | | 20.5 | 34.0 | 4.3 | 13.0 | 59.0 | |
| 60Fe-Co-K/40MMC | | | | 23.0 | 27.0 | 5.1 | 19.0 | 44.0 | |
| 80Fe-Co-K/20MMC | | | | 21.5 | 29.0 | 5.8 | 17.0 | 47.0 | |
| 40Fe-Co-K/60MC | | | | 23.0 | 59.0 | 2.5 | 7.0 | 37.0 | |
| 60Fe-Co-K/40MC | | | | 18.7 | 43.0 | 3.8 | 13.0 | 44.0 | |
| FeCo | 3:1 | 310 °C 2.0 MPa (GHSV) 1500 h$^{-1}$ | - | 28.0 | 46.4 | 16.8 | 20.9 | 15.9 | |
| | | | H$_2$ 450 °C 1 h. | 26.3 | 16.6 | 4.4 | 53.9 | 25.1 | |

However, Fe-Co/Al$_2$O$_3$ catalysts effectively increase the selectivity of methane formation during the hydrogenation of carbon dioxide ($CO_2$) to 87%. They also achieve high olefin content in C2+ hydrocarbons using potassium-promoted Fe–Co catalysts. Potassium-promoted Fe-Co catalysts also show better results in the synthesis of higher hydrocarbons, compared to other Fe/Al$_2$O$_3$, Co/Al$_2$O$_3$, and Fe-Co/Al$_2$O$_3$ catalysts.

It is worth noting that, in cases where there was preliminary activation with hydrogen, the distribution of products shifted towards the formation of methane. Without pre-activation, the main product was C2+ hydrocarbons.

At low temperatures, $CO_2$ hydrogenation without activation proceeds predominantly along the methanation pathway, which is comparable with the data given in [14,17,26], where a high methane selectivity (more than 60%) and $CO_2$ conversion (below 25%) are presented for iron–cobalt unpromoted $CO_2$ hydrogenation catalysts under various hydrogenation conditions, at temperatures below 350 °C.

It is also worth noting the close values of the specific activity for CO in both cases, which indicates the same adsorption activity of the contacts. However, the lower selectivity of CO formation during $CO_2$ hydrogenation without preliminary activation indicates the predominance of the first type of active sites, on which CO is dissociatively adsorbed and then hydrogenated to methane [27], and in the case of activation, the Fischer–Tropsch reaction predominantly proceeds on iron active sites.

Indeed, according to these authors, during $CO_2$ hydrogenation, the weak adsorption strength of $CO_2$ and the low partial pressure of CO are insufficient to inhibit H$_2$ adsorption, which leads to the active hydrogenation of CH$_3$* to methane and other low molecular

weight saturated hydrocarbons. As the $CO_2$ conversion increases, the yield and CO selectivity increase, due to the reverse water–gas shift reaction. High partial pressure and CO adsorption, in turn, slows down the hydrogenation of $CH_3^*$ to methane and leads to high values of the chain propagation probability, characteristic of the Fischer–Tropsch reaction path [28].

### 2.3. Physicochemical Results and Discussions

In the process of hydrogen treatment, the structure of the CoFe alloy is rearranged from the disordered state (date base PDF#49-1568) to the ordered state (date base PDF#49-1567), from which it can be assumed that, in ordered state, the surface of active centers contributes to a deeper flow water–gas shift reaction (Figure 5). The reverse reaction of the water shift reaction, occurring on the iron-containing fragments, produces carbon monoxide, which is sorbed on the cobalt-containing fragment of the particle, after which it is hydrogenated. This sequential process facilitates the water shift reverse reaction.

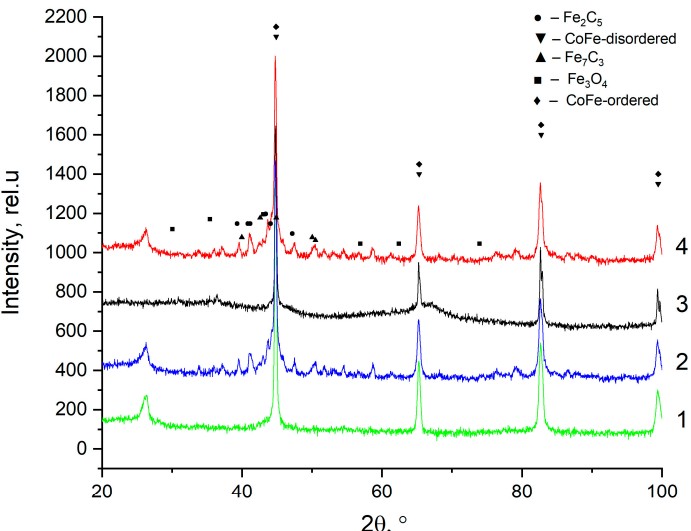

**Figure 5.** X-ray diffraction pattern of: 1—precursor sample after hydrogen treatment; 2—precursor sample after hydrogen treatment and catalytic experiment; 3—precursor sample; 4—precursor sample after catalytic experiment.

Structural studies were also carried out for samples after catalytic tests. The diffraction patterns had a similar appearance, and they recorded reflections of the carbide phase-$Fe_7C_3$ (date base JCPDS-75-1499) and $\chi$-$Fe_5C_2$ (date base JCPDS-51-0997) an insignificant amount of $Fe_3O_4$ (date base JCPDS-79-0419), and CoFe alloy. The carbide phase is the main active phase in the Fischer–Tropsch synthesis, from which it can be concluded that a polycondensation process took place on it with the formation of carbon-containing compounds with $C \geq 2$.

In the Raman spectra (Figure 6), distinct peaks around 1346 cm$^{-1}$ and 1586 cm$^{-1}$ belong to the D band of disordered graphitic carbon and the G band arising due to the stretching vibrations of the C–C bond in the graphite lattice plane. The intensity ratio of the D and G bands (ID/IG) may reflect the relative amounts of graphitic structure and disordered structure in the carbon components [29–31]. The ratio of band intensities for samples 1, 2, 3, and 4 is about 0.8, which indicates a high degree of ordering of graphitization, which increases during catalytic tests [32]. The D/G ratio decreases, indicating a greater degree of reduction and as a result of the formation of $sp_2$ clusters surrounded by an amorphous phase. Thus, we have two possible, but not mutually exclusive, effects with opposite contributions to the Raman band. On the one hand, we have reduction, the transformation of carbon $sp_3$ into $sp_2$; on the other hand, we have structural defects caused by gas evolution.

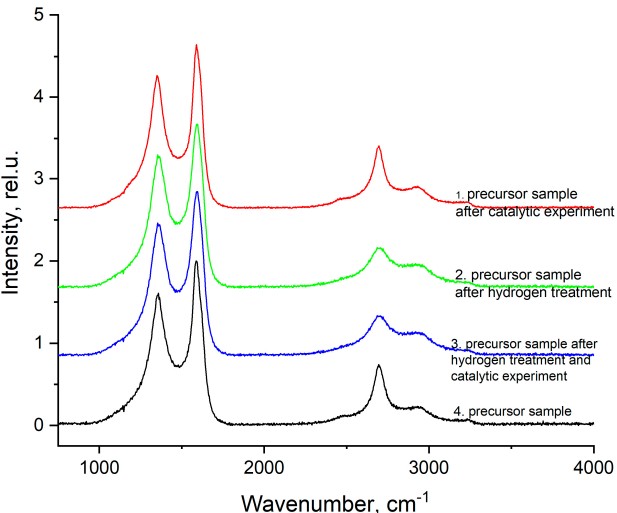

**Figure 6.** Ramanspectraof 1—precursor sample after catalytic experiment; 2—precursor sample after hydrogen treatment; 3—precursor sample after hydrogen treatment and catalytic experiment; 4—precursor sample.

In addition, the second-order bands located in the range from 2457 to 3000 cm$^{-1}$ are associated with a combination of modes/overtones, 2D, G′, and D + G, which may indicate the presence of single-walled carbon nanotubes and graphene with a small number of layers in the structure [33]. The presence of closed carbon nanotubes in the catalyst was previously confirmed in [19] using scanning electron microscope(SEM). Peaks around 3235 cm$^{-1}$ probably relate to symmetrical stretching vibrations of residual aliphatic nitriles (R–NH$_3$); the presence of a small amount of nitrogen is also confirmed by elemental analysis.

To understand the catalytic process, knowledge of the composition of the catalyst surface and its structure is important. The X-ray photoelectron spectroscopy (XPS) method was used to study the surface structure of the starting material, the material after surface treatment with hydrogen (activation), and the catalysts after the catalytic process (Table 4).

The surface of the material consists predominantly of carbon, which indicates the formation of carbon structures, consistent with Raman spectroscopy data and SEM data presented in [19]. Treatment with hydrogen leads to a decrease in nitrogen content, which may be due to the destruction of nitrogen-containing structures. Initially, the iron content on the surface exceeds the cobalt content, while the reduction treatment leads to a change in the ratio of metals on the surface. After the catalytic reaction, the ratio of metals is equalized.

For a more detailed study of the surface, detailed spectra of the elements were recorded (Table 5). From the deconvolution of the C1s spectrum, we can conclude that carbon is presented predominantly in the form of a sp$_2$-hybridized carbon phase [34] (284.4 eV, FWHM 1.3055), sp$_3$-hybridized carbon atoms and/or C–O, C–OH bonds [35] (285.6 eV, FWHM 1.57059) and C–O bonds [6] (286.6 eV, FWHM 4.62252). There are also signals from the $\pi \rightarrow \pi^*$ (291.5 eV, FWHM 5.0804) transition in graphene structures [36]. The deconvolution of O1s shows that oxygen on the surface is predominantly in the form of O–C–O [37] (533.3 eV, FWHM 3.54984), Me-O [38] (530.3 eV, FWHM 2.03983), and C=O groups, such as ketone and carbonyl [39] (531.8 eV, FWHM 1.40402). Nitrogen is represented by graphitic-N [40] (400.4 eV, FWHM 3.62972) and pyridinic-N [41] (398.0 eV, FWHM 2.07448). You can also separately identify a line with BE equal to 404.5 (FWHM 5.52943), which can be attributed to $\pi$ excitations, resulting in positive charge accumulation on the N species located at the edges [41]. The deconvolution of the Co2p spectrum shows that spinel CoFe$_2$O$_4$ (779.6 eV, FWHM 3.17869; 782.8 eV, FWHM 6.99552) is formed in the system [42,43].

**Table 4.** Composition of the surface of catalysts synthesized from DCS obtained by XPS.

| Sample | C, at. % | O, at. % | N, at. % | Fe, at. % | Co, at. % |
|---|---|---|---|---|---|
| precursor sample | 94.3 | 3.8 | 1.3 | 0.4 | 0.2 |
| precursor sample after hydrogen treatment | 92.3 | 6.8 | 0.1 | 0.3 | 0.5 |
| precursor sample after hydrogen treatment and catalytic experiment | 94.5 | 3.8 | 1.3 | 0.2 | 0.2 |
| precursor sample after catalytic experiment | 93.6 | 5.5 | 0.1 | 0.4 | 0.4 |

**Table 5.** Detailed element spectrum of samples and deconvolution.

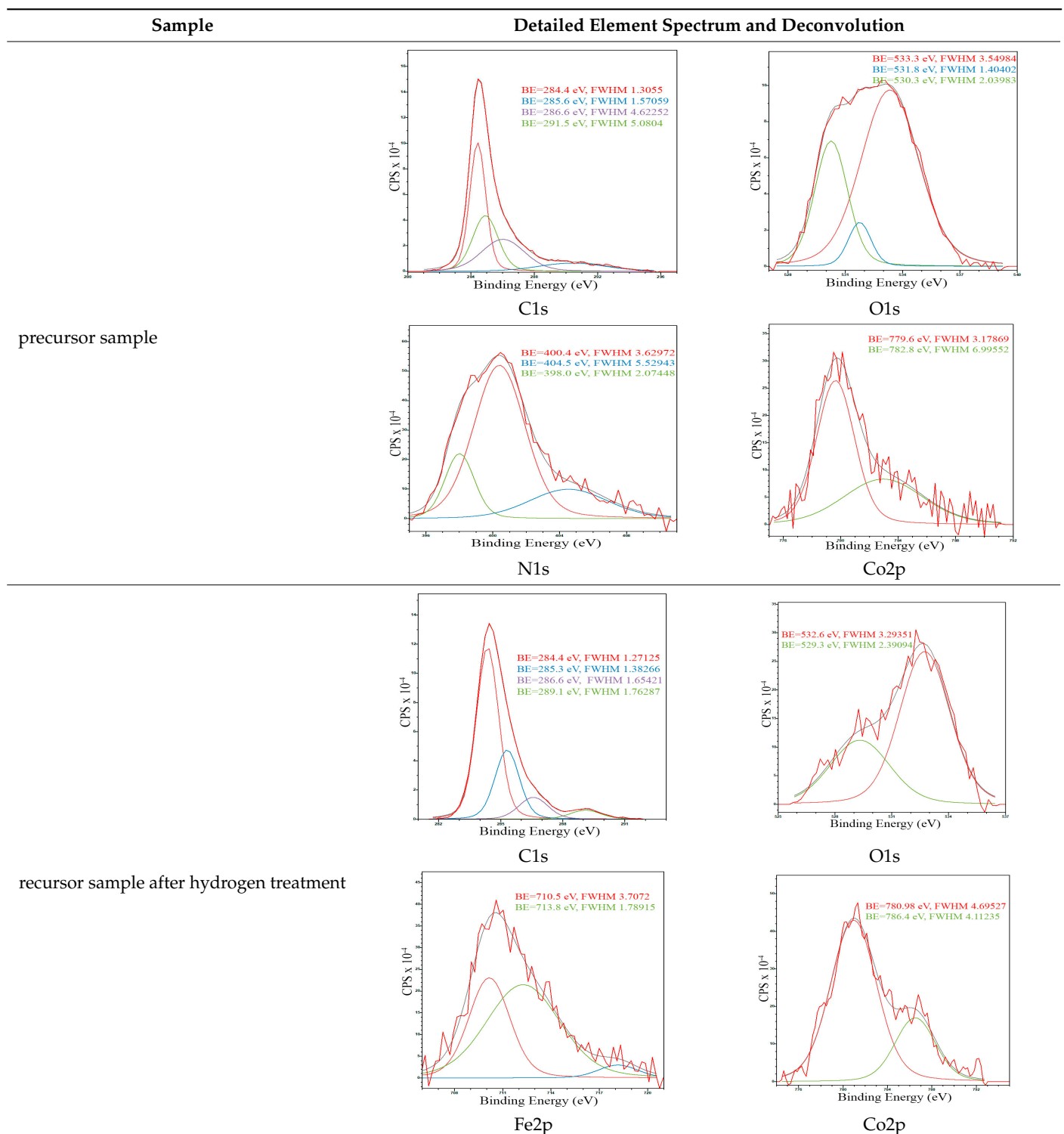

**Table 5.** *Cont.*

| Sample | Detailed Element Spectrum and Deconvolution |
|---|---|
| precursor sample after hydrogen treatment and catalytic experiment | 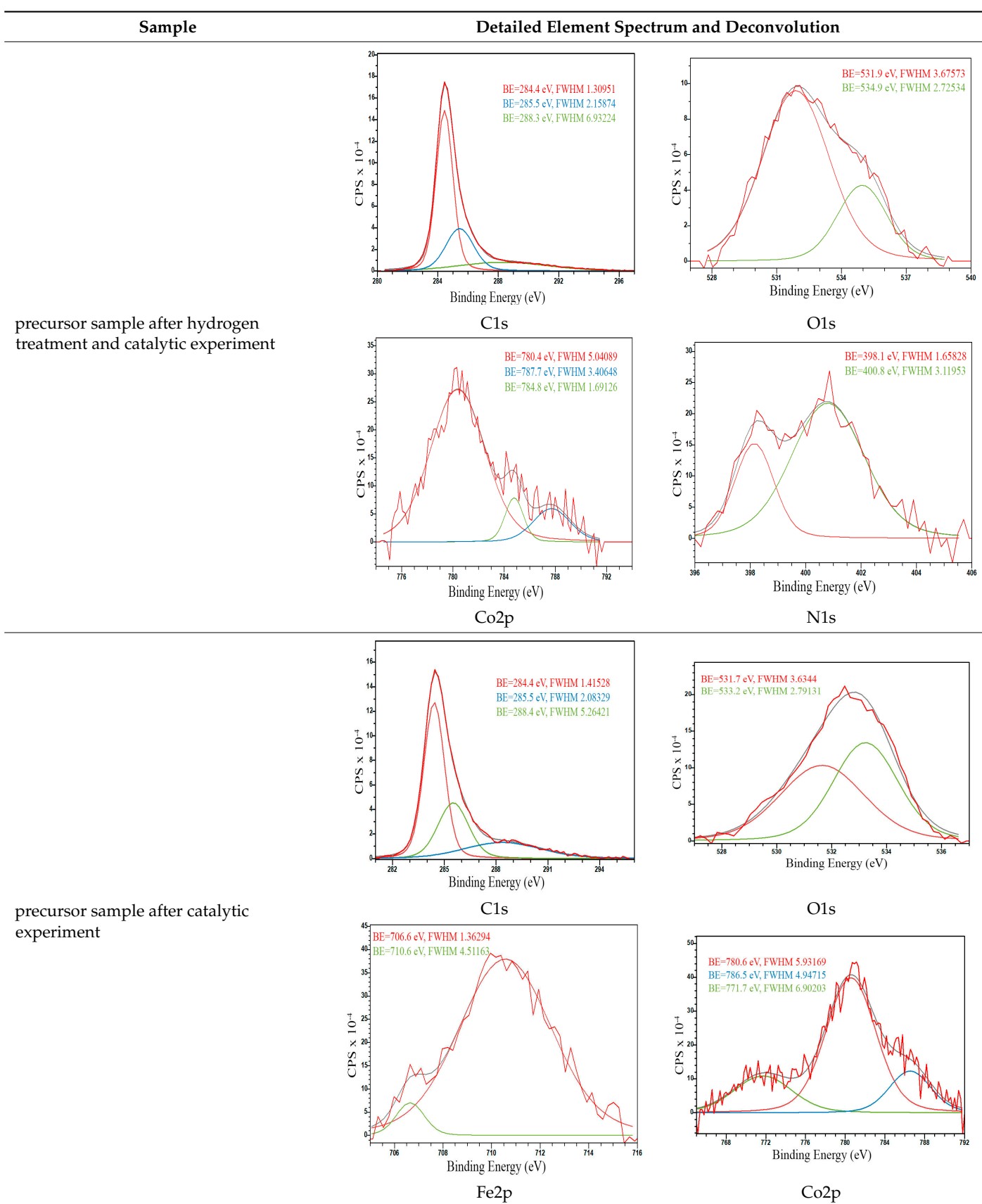 |
| precursor sample after catalytic experiment | |

After reduction, the surface composition undergoes changes; the deconvolution of the C1s spectrum, along with the $sp_2$-hybridized carbon phase [34] (284.4 eV, FWHM 1.27125), identifies the presence of the carboxylic group [44] (289.1 eV, FWHM 1.76287) and the C–N/C–O groups [45] (285.3 eV, FWHM 1.38266), C–O [46] (286.6 eV, FWHM 1.65421). Oxygen-containing groups are presented as epoxide C–O–C [47] (532.6 eV, FWHM 3.29351) and lattice oxygen [48] (529.3 eV, FWHM 2.39094). The deconvolution of the Fe2p spectrum shows the presence of $Fe^{2+}$ [49] (710.5 eV, FWHM 3.7072) and $Fe^{3+}$ [50] (713.8 eV, FWHM 1.78915). Cobalt is presented as $Co^{2+}$ [35] (781.0 eV, FWHM 4.69527).

Based on XPS data, it can be assumed that, at the stage of catalyst preparation, a structure is formed, with a certain metal core represented by the CoFe alloy, which is covered with the resulting carbon tube-like structures (see Raman spectra data). On the surface of the structures, there are heteroatoms (N, O), which can act as defects and interact with both the substrate and the active particle. However, for the most part, the carbon coating acts as a protective coating and prevents the oxidation of the alloy structure. The "unprotected place" of contact of the alloy with the environment is represented in the form of the spinel phase $CoFe_2O_4$. Thus, it can be assumed that the spinel structure exhibits insignificant activity compared to oxide structures in the process of $CO_2$ hydrogenation and CO formation; therefore, the conversion of raw materials at the initial stages (when the reaction temperature is 230–270 °C) is less than the conversion of $CO_2$ for the sample after the activation stage (Table 2). The spinel structure, when destroyed, forms nanoclusters of metallic cobalt and the oxide phase of iron. Thus, a bifunctional catalyst is formed, in which a reverse water shift reaction occurs on the oxide phase ($Fe_3O_4$), and the hydrogenation of carbon monoxide occurs on the formed metal cobalt particles ($Co^0$). It can be assumed that, under such conditions (smooth heating and holding in isothermal mode), smaller-sized cobalt particles are formed , which, according to [51], does not contribute to the growth of the carbon chain, but promotes the process of hydrogenation of carbon monoxide tomethane. During preliminary activation, the surface phase of spinel is destroyed and a bifunctional system is formed before the catalysis stage, which affects the initial activity exhibited by the catalyst during the hydrogenation process (Table 2). The resulting oxide structure ($Fe_3O_4$) promotes the reverse reaction of water–gas, and the formed cobalt clusters are larger, which promotes the process of carbon chain growth. In addition, it is worth noting the previously described (see X-ray diffraction data) formation of the iron carbide phase ($Fe_7C_3$ and $\chi$-$Fe_5C_2$), on which the hydrocarbon chain growth reaction can occur.

Thus, during catalytic tests without activation by hydrogen, the spinel structure formed on the surface promotes the occurrence of the reverse water shift reaction, and the Co phases, which are partially reduced during the synthesis process, promote the occurrence of methanation. In the case of a catalyst that has undergone pre-activation, the destruction of the spinel structure probably occurs with the formation of metal particles, which tend to both accelerate the reaction of the reverse water shift reaction and the process of hydrogenation and growth of the hydrocarbon chain.

The efficient conversion of $CO_2$ to $CH_4$ and CO provides an important opportunity to obtain valuable raw materials for various industrially important reactions, since both $CH_4$ and CO are widely used as starting materials for the synthesis of valuable fuels and chemicals. It is also important to note that it is possible to influence the selectivity of the carbon dioxide hydrogenation process in the presence of $[Co(NH_3)_6][Fe(CN)_6]$ catalysts using pre-treatment. This fact makes it possible to obtain, in addition to methane and carbon monoxide, other valuable products—$C_2$–$C_4$ hydrocarbons, which can be used as valuable petrochemical raw materials, and $C_{5+}$ hydrocarbons, which can be used as liquid fuel.

## 3. Materials and Methods

### 3.1. Materials

DCS $[Co(NH_3)_6][Fe(CN)_6]$ was obtained by mixing equivalent amounts of solutions of $[Co(NH_3)_6]Cl_3$ and $K_3[Fe(CN)_6]$. All reagents were purchased from Vecton. The elemental

analysis, IR spectroscopy, and powder X-ray diffraction (XRD) results are the same as described in [19]. KBr was used, with a purity of 99.9 wt. % (Neva-Reaktiv).

### 3.2. Obtaining Catalytic Composition

The catalyst was the product of the thermal destruction of $[Co(NH_3)_6][Fe(CN)_6]$ in argon (99.999 weight %) at 650 °C within 1 h, at a heating rate of 10 °C/min. The tubular muffle furnace Nabertherm RT 50-250/11 (Nabertherm GmbH, Lilienthal, Germany, 2013) was used.

### 3.3. Physicochemical Research Methods

Elemental analysis was carried out using the analyzer ELTRA-2000 (Alpha Resources, LLC, Stevensville, MI, USA, 2004) and on an atomic absorption spectrometer with a hydride attachment, and a flow-through sample preparation unit for sorption concentration "Kvant-2A" (LLC Kortec, 2003, St. Petersburg, Russia).

Fourier transform infrared spectra were recorded with a Nicolet 6700 FT-IR spectrophotometer (Thermo Fisher Scientific Inc., Hillsboro, OR, USA, 2010) in the wavelength range of 400 $cm^{-1}$ to 4000 $cm^{-1}$, with tablet KBr, 16 scans, 4 resolution $cm^{-1}$.

Analysis N was conducted by chromatography of the oxide mixtures obtained after sample combustion with oxygen in a dynamic flash at $\approx$2000 °C, using a Thermo Flash 2000 analyzer (Thermo Fisher Scientific, Heysham, UK). The detector was a katharometer, and the carrier gas was helium.

The porous structure of the samples was studied using the method of low-temperature sorption of nitrogen on a Tristar 3020 instrument (Micrometritics, Norcross, GA, USA, 2009). The X-ray phase analysis of the products of the thermal destruction of DCS was made in 2θ range 10–100° on a Shimadzu XRD 6000 powder diffractometer (Shimadzu, Kyoto, Japan, 2008), equipped with a Cu-Ka source (λ = 1.5418 Å) and a graphite monochromator for the diffracted beam. Indexing of the diffraction patterns was performed using the data for pure metals and compounds reported in the JCPDS-ICDD PDF4+ database (2019).

The analysis of the products was carried out on a Khromos GC-1000 gas-adsorption chromatographic complex. The detector was a katharometer, and the carrier gas was helium with 5% nitrogen as an internal standard, which was supplied at a flow rate of 20 mL/min. Two columns were used for the analysis: the first column separated CO, $N_2$, and $CH_4$ gases on CaA molecular sieves under isothermal conditions at a temperature of 80 °C, and the second column served to separate $CO_2$ and $C_2$–$C_4$ hydrocarbons in the temperature range of 80–200 °C and was filled with HayeSep R.

The Raman spectra were obtained using a confocal Raman microscope Senterra II (Bruker, Billerica, MA, USA). A laser with a wavelength of 532 nm and a power of 0.25 mW was used to excite the Raman scattering. The accumulation time was 1 s, the number of repetitions was 200, the objective was 50X, the diffraction grating had 400 lines/mm, the resolution was 4 $cm^{-1}$, and the aperture was 50 × 1000 μm. Ten spectra from different selected areas were recorded for each sample. Spectral processing was carried out using the OPUS 8.5 software package (Bruker, Billerica, MA, USA).

The study of the composite material samples's urface was carried out by X-ray photoelectron spectroscopy on an X-ray photoelectron spectrometer (Prevac, Rogow, Poland). An X-ray tube with AlK$\alpha$ radiation (1486.6 eV) was used as a source of ionizing radiation. Before being loaded into the spectrometer, the samples were ground in an agate mortar and applied to conductive carbon tape. To neutralize the charge of the sample during the experiments, an electron-ion charge compensation system was used. All peaks were calibrated versus the C 1s peak at 284.8 eV. The type of background was Shirley and during deconvolution, it was assumed that the total peak was the sum of Gaussian curves.

### 3.4. Simultaneous Thermal Analysis

Simultaneous thermal analysis was carried out by using an STA409 PC/PG microthermoanalyzer (Netzsch, Selb, Germany). All studies were carried out in an argon at-

mosphere with a flow rate of 75 mL/min and heating rates of 1, 10, and 20 °C/min. The measurements were taken in the temperature range from 40 to 1000 °C. Samples weighing 11–18 mg were placed in $Al_2O_3$ crucibles (80 μL), with lids that had holes of 0.5 mm diameter, for experiments.

*3.5. Kinetic Analysis*

The kinetic parameters of the thermolysis process were calculated by using the methods of non-isothermal kinetics: by using isoconversion and model approaches. The conversion was readily determined as a fractional change in any physical property associated with the reaction progress. When the process progress was monitored as a change in mass by TGA, $\alpha$ was determined as a ratio of the current mass change, $\Delta m$, to the total mass change, $\Delta m_{tot}$, which occurred throughout the process:

$$a = \frac{m_0 - m}{m_0 - m_f} = \frac{\Delta m}{\Delta m_{tot}} \tag{1}$$

where $m_0$ and $m_f$, respectively, are the initial and final masses. The general form of the basic rate equation is usually written as:

$$\frac{d\alpha}{dt} = k(T)f(a) \tag{2}$$

where $T$ is the temperature, $f(a)$ is the differential form of the reaction model that represents the reaction mechanism, and $k(T)$ is the rate constant. The dependence of the rate constant on temperature is given by the Arrhenius law:

$$\frac{d\alpha}{dt} = A exp\left(\frac{-E}{RT}\right)f(a) \tag{3}$$

where $A$ is the pre-exponential factor (in $s^{-1}$). Isoconversion approaches are based on the assumption that the reaction rate depends only on temperature [52]. According to Equation (2):

$$\left[\frac{\partial ln(da/dt)}{\partial T^{-1}}\right]_a = \left[\frac{\partial lnk(T)}{\partial T^{-1}}\right]_a + \left[\frac{\partial lnf(a)}{\partial T^{-1}}\right]_a \tag{4}$$

the subscript a indicates the values related to a given extent of conversion. Since $f(a)$ does not depend on $T$, when a is constant, Equation (4) reduces to:

$$\left[\frac{\partial ln(da/dt)}{\partial T^{-1}}\right]_a = -\frac{E_a}{R} \tag{5}$$

A model-free value of the apparent activation energy $E_a$ can thus be estimated for each a value from Equation (5). As a result, the value of $E_a$ is a function of $a$. By rearranging Equation (3), one can derive the basic equation of the Friedman method [53]:

$$ln\left(\frac{da}{dt}\right)_{a,i} = ln[f(a)A_a] - \frac{E_a}{RT_{a,i}} \tag{6}$$

Integral isoconversional methods originate from the application of the isoconversional principle to the integral in Equation (3). The integral in Equation (3) does not have an analytical solution for an arbitrary temperature program. Starink [54] demonstrated that many approximations give rise to linear equations of the general form:

$$ln\left(\frac{\beta_i}{T_{a,i}^B}\right) = Const - C\left(\frac{E_a}{RT_a}\right) \tag{7}$$

The crude temperature integral approximation results in inaccurate values of $E_a$. A more accurate approximation by Murray and White gives rise to B = 2 and C = 1, and leads to the popular equation that is frequently called the Kissinger–Akahira–Sunose (KAS) equation [55]:

$$ln\left(\frac{\beta_i}{T_{a,i}^2}\right) = Const - C\left(\frac{E_a}{RT_a}\right) \tag{8}$$

Kinetic parameters were calculated by using both model-based and isoconversional approaches. The Friedman analysis and the KAS method were used to calculate the activation energy of thermolysis in this study. Model methods were based on the determination of kinetic parameters by minimizing the difference between experimentally measured and calculated data. The evaluated models include n-dimensional nucleation, according to Avrami–Erofeev (An), reaction of the nth order (Fn), reaction of the nth order with m-Power autocatalysis by product (Cnm), expanded Prout–Tompkins equation (Bna), and the expanded Sestak–Berggren equation (Sb). The equations for the models are collected in Table 6.

**Table 6.** Models' methods for calculating kinetic parameters.

| Model | Equation |
|---|---|
| Fn | $f = (1 - \alpha)^n$ |
| An | $f = n \times (1 - \alpha) * [-ln(1 - \alpha)]^{\left(\frac{n-1}{n}\right)}$ |
| Bna | $f = (1 - \alpha)^n * \alpha^{AutocatOrder}$ |
| Cnm | $f = (1 - \alpha)^n * (1 + AutocatPreExp * \alpha^m)$ |
| Sb | $f = (1 - \alpha)^n * \alpha^m * [-ln(1 - a)]^q$ |

The kinetic parameters of thermolysis were determined based on the different heating rates. Kinetic analysis was performed, according to the ICTAC recommendations [56–58], by using the NETZSCH Kinetics Neo 2.6.6.7 software package.

*3.6. Catalytic Tests*

Catalytic tests were carried out in the temperature range from 230 °C to 310 °C, with a step of 20 °C and exposure for 12 h for each temperature, at a pressure of 2.0 MPa in a flow unit with a fixed catalyst bed, gaseous reagents ($H_2:CO_2$ = 3:1) and space velocity (GHSV = 1500 h$^{-1}$). Catalytic tests were carried out at the stage of preliminary activation of the sample with hydrogen at a temperature of 450 °C for 1 h, a pressure of 2.0 MPa, and the space velocity of $H_2$ (GHSV = 1000 h$^{-1}$) and without the stage of preliminary activation. After activation, the reactor was cooled to 230 °C and gaseous reactants ($H_2:CO_2$ = 3:1) were fed into the reactor at 2.0 MPa and space velocity (GHSV = 1500 h$^{-1}$).

The efficiency of the tested system was evaluated from the calculation of output indicators: $CO_2$ conversion, selectivity, and the specific activity of the catalyst.

$CO_2$ conversion is the ratio of the mass of reacted $CO_2$ to the mass of carbon dioxide that entered in the reaction zone (9):

$$X_{CO_2} = \frac{M_{CO_2}^{reacted}}{M_{CO_2}^{entered}} \times 100\% \tag{9}$$

The selectivity of the formation of products is the ratio of the amount of product formed to the amount of reacted raw materials (10):

$$S = \frac{M_{products}}{M_{CO_2}^{reacted}} \times 100\% \tag{10}$$

The specific MTY of a catalyst is the number of reacted moles of $CO_2$ per gram of Fe-Co per second:

$$MTY = \frac{N_{CO_2}}{M_{Fe-Co} \cdot t} \tag{11}$$

## 4. Conclusions

The process of carbon dioxide hydrogenation in the presence of a bimetallic bifunctional catalyst, based on a system obtained by the thermolysis of $[Co(NH_3)_6][Fe(CN)_6]$, was studied. The catalyst formation process was studied using kinetic methods. The kinetic parameters of the $[Co(NH_3)_6][Fe(CN)_6]$ thermolysis process were determined by thermogravimetry using isoconversion and non-isothermal kinetics model approaches. All stages of the thermolysis process are well described by models of autocatalytic processes. The activation energies determined by different approaches have similar values. This indicates the reliability of the obtained results. It is shown that, during the preparation process, composite materials are formed, which are as follows: a core CoFe alloy, a carbon coat based on a carbon nanotube-like structure, and a metal-containing component represented by spinel $CoFe_2O_4$. The resulting catalysts represent a new class of catalysts that can be used without the entire pre-activation step. Separately, we can note the possibility of controlling the selectivity of the process by treating the catalyst with hydrogen. Further study of the effect on the selectivity of the hydrogenation process of carbon dioxide, in the presence of catalysts based on thermolyzed double complex salts, can be associated with the study of the influence of promoting additives. It is also advisable to establish the influence of the conditions of various activation processes' changes in temperatures, pressures, flows, and composition of the activation raw material.

**Supplementary Materials:** The following supporting information can be downloaded at: https://www.mdpi.com/article/10.3390/catal13121475/s1. Table S1. Kinetic parameters of the first stage of the $[Co(NH_3)_6][Fe(CN)_6]$ thermolysis process. They were determined using model methods. Table S2. Kinetic parameters of the second stage of the $[Co(NH_3)_6][Fe(CN)_6]$ thermolysis process. They were determined using model methods.

**Author Contributions:** Conceptualization, Y.P.S.; investigation, A.N.G., M.V.K., Y.P.S., N.S.T., M.I.I., A.A.G., S.E.L. and A.V.G.; methodology, A.N.G., M.V.K. and Y.P.S.; project administration, A.N.G.; writing—original draft, A.N.G., M.V.K. and S.E.L.; writing—review and editing, A.N.G., M.V.K., Y.P.S., N.S.T., M.I.I., A.A.G., S.E.L. and A.V.G. All authors have read and agreed to the published version of the manuscript.

**Funding:** This work was financially supported by the grant for young scientists of the Murmansk region (Russia) № 134 of 3 May 2023, and has been carried out according to the framework of Scientific Research Contracts, Russian Federation, No. FMEZ-2022-0017 and the State Program of TIPS RAS Russian Federation.

**Data Availability Statement:** The data presented in this study are available.

**Conflicts of Interest:** The authors declare no conflict of interest.

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
