# Peer review of "CO2 Hydrogenation over Fe-Co Bimetallic Catalyst Derived from the Thermolysis of [Co(NH3)6][Fe(CN)6]"

_catalysts, doi:10.3390/catal13121475_

Round 1

Reviewer 1 Report

Comments and Suggestions for Authors

In this work, the authors reported the synthesis of a new type of bimetallic catalyst by a simple thermolysis method for the selectivity of the carbon dioxide hydrogenation. Various physicochemical methods - XRD, IR-spectroscopy, Raman spectroscopy, AND XPS were employed to characterize the catalysts. Besides, this paper was well structured and written. Therefore, I would like to recommend its acceptance after the following revisions.

1.       The authors are suggested to provide more discussion on the origin of the catalytic activity by analyzing the changes in the valence states of the catalyst.

2.       The origin of the performances should be analyzed in more detail.

3.       What are the roles of metal atoms in improving catalytic performance?

4.       To make the paper more compressive and convincing, critical references reflecting the recent advances of catalysts are suggested to be cited, for example, Angew. Chem. Int. Ed. 2023, 62, e202215968;  Chemical Research in Chinese Universities, 2021, 37, 1268

Author Response

â„–

Comment

Answer

1

The authors are suggested to provide more discussion on the origin of the catalytic activity by analyzing the changes in the valence states of the catalyst.

We agree with the comment. More detailed discussions have been included in the text of the article..

2

The origin of the performances should be analyzed in more detail.

We agree with the comment. More detailed discussions have been included in the text of the article.

3

What are the roles of metal atoms in improving catalytic performance?

We agree with the comment. More detailed discussions have been included in the text of the article.

4

To make the paper more compressive and convincing, critical references reflecting the recent advances of catalysts are suggested to be cited, for example, Angew. Chem. Int. Ed. 2023, 62, e202215968; Chemical Research in Chinese Universities, 2021, 37, 1268

We agree with the comment. More detailed discussions have been included in the text of the article.

Reviewer 2 Report

Comments and Suggestions for Authors

In this manuscript, the authors report the use of bimetallic (Co-Fe) salts for CO2 hydrogenation to different C-based species. The effect of activation protocol has been analyzed on catalytic activity and selectivity. To gain insight on observed differences, non-activated and preactivated catalysts are characterized by XRD, IR-spectroscopy, XPS and thermogravimetric analysis. However, the evidence for some of the conclusions in the manuscript is insufficient and there are still some questions to be answered.

1.       In general, there are many grammatical mistakes and the English level should be improved.

2.       The abstract should be revised. Complex and not clear. In the abstract, striking sentences emphasizing the work should be added. In addition, the numerical data obtained in the study should be mentioned.

3.       Considering the quality of the journal, the introduction should be more striking and rewritten. In fact, the introduction is too long, includes a mix a very different concepts and does not include a clear explanation of the state of the art, the starting hypothesis and the main goal of this study.

4.       TG curves are not described and discussed.

5.       It is not clear which is the main purpose of the thermokinetic study. Why the main conclusion extracted from this study are useful to explain the rest of the results?

6.       The decimal separator in Table should be corrected.

7.       A comparison table should be given with similar studies in the literature, especially in terms of catalytic activity.

8.       Deconvoluted XPS spectra should also be included in order to provide evidenced about the differences claimed during the text.

9.       It is not clear which are the main differences in the physic-chemical properties between activated and non-activated samples that allow explaining the observed differences in CO2 conversion and selectivity. A more in-depth discussion should be done by linking the results obtained from characterization and activity test.

Comments on the Quality of English Language

Moderate editing of English language is required

Author Response

â„–

Comment

Answer

1.

In general, there are many grammatical mistakes and the English level should be improved.

We agree with the comment.

2.

The abstract should be revised. Complex and not clear. In the abstract, striking sentences emphasizing the work should be added. In addition, the numerical data obtained in the study should be mentioned.

We agree with the comment. Abstract updated.

3.

Considering the quality of the journal, the introduction should be more striking and rewritten. In fact, the introduction is too long, includes a mix a very different concepts and does not include a clear explanation of the state of the art, the starting hypothesis and the main goal of this study.

We agree with the comment. The introduction has been updated.

4.

TG curves are not described and discussed.

We agree with the comment. Corrections have been made to the text of the article.

5.

It is not clear which is the main purpose of the thermokinetic study. Why the main conclusion extracted from this study are useful to explain the rest of the results?

Because The final product of the decomposition of the initial complex has catalytic properties; it is necessary to know the kinetic parameters of this process, which will allow the selection of optimal heating modes for the formation of the catalyst.

6.

The decimal separator in Table should be corrected.

We agree with the comment. Corrections have been made to the text of the article.

7.

A comparison table should be given with similar studies in the literature, especially in terms of catalytic activity.

We agree with the comment. Corrections have been made to the text of the article.

8.

Deconvoluted XPS spectra should also be included in order to provide evidenced about the differences claimed during the text.

We agree with the comment. Corrections have been made to the text of the article.

9.

It is not clear which are the main differences in the physic-chemical properties between activated and non-activated samples that allow explaining the observed differences in CO2 conversion and selectivity. A more in-depth discussion should be done by linking the results obtained from characterization and activity test.

We agree with the comment. Corrections have been made to the text of the article.

Reviewer 3 Report

Comments and Suggestions for Authors

In this paper, the topic is clear, and the research method is rigorous. The improvement of CO2 hydrogenation performance of Fe-Co bimetallic catalyst by pretreatment method is studied, and the kinetic parameters of catalytic conversion reaction are calculated. This paper provides a reference for developing high-efficiency catalysts for CO2 catalytic hydrogenation. The work is suitable for publication in the Journal of Catalysts. However, this article should be slightly revised before publication.

1. Some contents discussed in the introduction were not highly relevant to the theme of this paper; the authors were suggested to revise this part carefully.

2. 2.2 Section title may be “Catalytic tests.”

3. All Figures(except Fig.1) should be revised; lines of different samples were suggested to be modified with other colors in these figures.

4.  XPS results were suggested to be added in this paper.

        5. The selective discussion does not specify which hydrocarbon is the final product, which lacks practical application.

Comments on the Quality of English Language

Minor editing of English language required

Author Response

â„–

Comment

Answer

1.

Some contents discussed in the introduction were not highly relevant to the theme of this paper; the authors were suggested to revise this part carefully.

We agree with the comment. Corrections have been made to the text of the article.

2.

2.2 Section title may be “Catalytic tests.”

We agree with the comment. Corrections have been made to the text of the article.

3.

All Figures(except Fig.1) should be revised; lines of different samples were suggested to be modified with other colors in these figures.

We agree with the comment. Corrections have been made to the text of the article.

4.

XPS results were suggested to be added in this paper.

We agree with the comment. Corrections have been made to the text of the article.

5.

The selective discussion does not specify which hydrocarbon is the final product, which lacks practical application.

We agree with the comment. More detailed discussions have been included in the text of the article..

Round 2

Reviewer 2 Report

Comments and Suggestions for Authors

The authors have fully addressed the previous comments and revised their manuscript accordingly. Thus, the current manuscript is recommended to be accepted now.

Comments on the Quality of English Language

English has been improved. However, some grammatical errors are still present.